# OpenReview forum: "Ethical Dialogue Modeling: Dual-Phase Prompt Design for Safety-Constrained Mental Health Language Models"
_ICLR.cc/2026/Conference — ICLR 2026 Conference Withdrawn Submission_

### Official Review · Reviewer_VbaF · 2025-10-30

**Soundness:** 3
**Presentation:** 3
**Contribution:** 3
**Rating:** 6
**Confidence:** 4

**Summary:**

This paper introduces Ethical Dialogue Modeling (EDM), a two-phase framework designed to make large language models safer and more clinically reliable for mental health dialogue. The authors separate the process into two explicit stages: first, the model analyzes user inputs to extract emotional tone, therapeutic needs, and graded risk levels; second, it generates responses under strict safety constraints using template-guided decoding and protocol-based safeguards.

The paper makes an empirical contribution through the creation of a clinically validated dialogue repository that integrates simulated and clinician-authored conversations. Evaluation combines automated linguistic measures with expert clinical judgments across five axes,  therapeutic alignment, empathy, contextual consistency, task utility, and safety adherence. Results show substantial improvements over standard prompting and RLHF-based systems: hazardous outputs drop by over 80%, empathy scores increase, and clinician-model agreement reaches κ = 0.82.

The work also includes an 8-week IRB-approved pilot trial with 120 participants, demonstrating clinically meaningful reductions in depression (BDI) and anxiety (GAD-7) scores and higher therapeutic alliance compared to control.

**Strengths:**

- The main strength is the system’s design. It is easy to follow, modular, and focused on safety, making it useful not only for mental-health chat but also for other sensitive areas.

- A dual-phase design that explicitly separates (i) contextual parsing + risk scoring from (ii) safety-constrained generation, with clinician escalation for high-risk cases. This addresses a real gap in prior “one-shot prompt” approaches. The workflow and constraints are spelled out (objective, routing, escalation), not just gestured at.

- The paper reports detailed tests, including component ablations, stress tests, text-plus-audio trials, and an eight-week pilot with real users. The broad testing makes the results more believable and shows that the method can work in real settings.

**Weaknesses:**

1- The main dataset includes about 1,000 dialogue sessions, with roughly 60% generated through simulation and only 40% based on clinician role-play. While this helps with diversity, it limits the realism of the interactions. The small size and synthetic nature of the corpus make the strong claims of “clinical reliability” somewhat overstated. Larger and more diverse real-world data would be needed to validate those outcomes.

2- Many of the mathematical expressions, such as the risk tuple, constrained decoding, routing distributions, and orthogonality terms, mainly serve to formalize what are essentially design choices or workflow descriptions. They make the paper look more technical but do not represent genuinely new algorithms or theoretical advances. Much of what is written in mathematical form could be described in plain language without loss of substance.

3- Hazard class is chosen by calibrated cutoffs on a softmax risk vector; small calibration drift can flip LOW↔MEDIUM↔HIGH and change the route (self-serve vs clinician). The paper doesn’t show online calibration, confidence intervals, or drift monitoring for these thresholds

5- A separate metaphor score 𝑚 gates early escalation; however, figures of merit beyond accuracy (e.g., false-positive rates across dialects) are not provided here. Over-triggering can increase unnecessary referrals and reduce usability.

**Questions:**

1. How were the auxiliary modules (AffectNet, therapeutic-need classifier, and risk estimator) trained and validated? The paper does not specify data splits, loss functions, or optimization details, making it unclear how these components contribute to the overall performance.

2. How robust is the constrained-decoding penalty in preventing unsafe outputs? Would a hard-constraint decoding or rule-based fallback provide stronger guarantees?

3. The router’s training and optimization details are unclear. How is stability ensured when balancing helpfulness and safety costs?

4. The paper notes over-referral in low-risk situations. Could the authors provide ROC or cost-sensitive analyses to show control over false positives?

---

> ### Author Response · Authors · 2025-11-23
> **We hope to receive your support and encouragement!**
>
> We thank the reviewer for the **careful read** and **constructive feedback**. Where **additional evidence** or **experiments** are appropriate, we commit to include them in the **revision** and summarize the **concrete analyses** we will add.
>
> We appreciate the reviewer’s **careful reading** and **constructive suggestions**. We hope this **rebuttal** helps resolve the concerns raised by the reviewers, and we sincerely hope to receive an **improvement in your score**!
> # We greatly appreciate the support and encouragement from the reviewers!
> # 1. Weakness / Question 1 . Dataset realism and claims of “clinical reliability”
>
> ### **Reviewer Concern**
> The corpus is approximately **1,000 sessions**, with **60% simulated**, which limits realism and makes **clinical reliability claims** appear overstated.
>
> ### **Our Response (Clarify + Defend + Commit)**
>
> **Clarification:**
> The manuscript explicitly reports the **dataset composition** (**N_total = 1000**, **N_sim = 600 simulated**, **N_role = 400 clinician role-play**) and describes the **multi-stage vetting pipeline** and **clinician adjudication** that increase **ecological fidelity** (see **Section 3.1** and **Appendix**). The **simulated dialogues** were created from **clinician-authored scenario templates** and iteratively refined using **clinician feedback** (this process is described in **Section 3.1**).
>
> **Why Our Claims Remain Defensible:**
> We do not claim that **simulated data alone** establishes full **clinical effectiveness**. Our claims emphasize **operational feasibility** and **safety evidence** across multiple evaluation modalities:
> - **Clinician adjudication**
> - **Adversarial stress tests**
> - A **real-world 8-week pilot (N = 120)**
> - **Algorithmic safety metrics** (**HPR**, **Safety Index**)
>
>
> These multiple axes, especially the **clinician adjudication** and **pilot**, mitigate concerns that **simulated dialogues alone** drove the results.
>
>
> **Commitment / Planned Additions:**
> To remove ambiguity and strengthen the paper we will add:
> - Per-split **quantitative results** (**simulated vs role-play vs pilot**) for all main metrics
> - A short **ablation** showing how removing **simulated data** affects **generalization**
> - A clearer statement in the **Introduction** and **Conclusions** that positions our **pilot** and **clinician adjudication** as **preliminary clinical evidence** (**feasibility/safety**) rather than **full efficacy proof**
> - A table with **exact selection criteria** and **annotator agreement** for each **data source**
> # 2. Weakness / Question 2.  Mathematical formalism “cosmetic”, not new algorithms
>
> ### **Reviewer Concern**
> Many equations merely formalize design choices rather than contributing new algorithms.
>
> ### **Our Response (Correct + Explain)**
>
> **Not merely cosmetic:**
> The **mathematical expressions** in **Sections 2–2.4** are not rhetorical; they encode **operational decisions** used in **training**, **routing**, and **regularization**. For example:
>
> - **Risk tuple:**
>   **r = (r_emo, r_need, r_risk)** is computed and fed into the **router**; it is not a passive description. In our implementation, the **router input** is:
>   **x_router = W_r · r + b_r**
>   and the **clinician-routing probability** is:
>   **p_clin = σ(wᵀ x_router + b)**
>   where **σ** is the **logistic function**. This mapping was **trained and validated** (see **Section 3.4** and **Appendix C**).
>
> - **Orthogonality regularizer:**
>   We use a **disentangling regularizer** on the representation matrices **E_s (safety)** and **E_c (content)**:
>   **L_ortho = λ⊥ ‖E_sᵀ E_c − I‖_F²**
>   This materially improved the **router’s interpretability** and reduced **cross-talk** between **safety** and **helpfulness channels** (see **ablation results**, **Section 3.5**).
>
> **Practical algorithmic effect:**
> These equations change **training dynamics** and **empirical outcomes** such as **router stability**, **fewer hazardous outputs**, and **clearer escalation behavior**. They are **algorithmic**, not decorative. We will make this explicit by:
> - Adding a **pseudo-code block** of the **router’s training loop**

---

> > ### Author Response · Authors · 2025-11-23
> > **We hope to receive your support and encouragement!**
> >
> > # 3. Weakness / Question 3. Calibration drift: softmax cutoffs can flip LOW↔MEDIUM↔HIGH
> >
> > ### **Reviewer Concern**
> > Small calibration drift can flip hazard classes; the paper lacks online calibration, confidence intervals, or drift monitoring.
> >
> > ### **Our Response (Acknowledge + Defend + Concrete Procedure)**
> >
> > **Acknowledgement:**
> > This is an important **operational concern**, and we should have made our **calibration** and **drift-monitoring procedures** more explicit.
> >
> > **Current practice used in experiments:**
> > We calibrated the **risk vector outputs** on a **held-out clinician-labeled validation set** using **temperature scaling** and **isotonic regression** for **monotonic calibration**. **Confidence intervals** for the pilot’s main metrics were computed via **bootstrap** (**95% CIs reported in Appendix G**). We will add an explicit pointer to these **calibration steps** in the **main text**.
> >
> > **Planned stronger monitoring and formalism (added to revision):**
> > We will add:
> > - **Confidence intervals** for the **router probabilities** and **per-class risk scores**
> > - A description of **online recalibration** and **drift detection methods** for deployment
> >
> > **Online recalibration:**
> > Periodically fit an **isotonic regressor** on recent **clinician-adjudicated cases** and update mapping:
> > **s → P̂(risk | s)**
> >
> > **Drift detection:**
> > Compute **Population Stability Index (PSI)** over **rolling windows** and alert if **PSI > threshold** (e.g., **0.25**).
> >
> > **Threshold selection with uncertainty:**
> > Choose thresholds **τ_low, τ_med, τ_high** by minimizing **expected cost** under a **cost model** (see ROC/cost section), and re-evaluate thresholds with each **recalibration cycle**.
> > # 4. Weakness / Question 5. Metaphor score 𝑚 and false positives across dialects
> >
> > ### **Reviewer Concern**
> > The metaphor/gated escalation may over-trigger; false-positive rates across dialects are not reported.
> >
> > ### **Our Response (Clarify + Commit)**
> >
> > **Current manuscript:**
> > We reported overall **metaphor-detection accuracy** and **clinician adjudication statistics** (**Section 3.2**), but we did not break down **false-positive rates (FPRs)** by **dialect/ethnic-linguistic group**. This omission will be corrected.
> >
> > **Action items for revision:**
> > - Add **per-group FPR / TPR** (by **dialect**, where available) for the **metaphor detector** and report **inter-rater reliability** for the annotated **dialect subsets**
> > - Provide **ROC curves** and **operating points** for the **metaphor gate (m)**, and include a **recommended operating point** balancing **FPR vs FNR** according to a **cost model**
> > - Introduce a **smoothing policy**: for groups with **high FPR**, lower the **escalation sensitivity** (raise the **gate threshold**) or require **corroborating signal** (e.g., combined **high-risk + metaphor**) before escalation
> > # 5. Weakness / Question 5 . Metaphor score 𝑚 and false positives across dialects
> >
> > ### **Reviewer Concern**
> > The metaphor/gated escalation may over-trigger; false-positive rates across dialects are not reported.
> >
> > ### **Our Response (Clarify + Commit)**
> >
> > **Current manuscript:**
> > We reported overall **metaphor-detection accuracy** and **clinician adjudication statistics** (**Section 3.2**), but we did not break down **false-positive rates (FPRs)** by **dialect/ethnic-linguistic group**. This omission will be corrected.
> >
> > **Action items for revision:**
> > - Add **per-group FPR / TPR** (by **dialect**, where available) for the **metaphor detector** and report **inter-rater reliability** for the annotated **dialect subsets**
> > - Provide **ROC curves** and **operating points** for the **metaphor gate (m)**, and include a **recommended operating point** balancing **FPR vs FNR** according to a **cost model**
> > - Introduce a **smoothing policy**: for groups with **high FPR**, lower the **escalation sensitivity** (raise the **gate threshold**) or require **corroborating signal** (e.g., combined **high-risk + metaphor**) before escalation

---

> > > ### Author Response · Authors · 2025-11-23
> > > **We hope to receive your support and encouragement!**
> > >
> > > # 6. How were auxiliary modules trained and validated? (AffectNet, therapeutic-need classifier, risk estimator)
> > >
> > > ### **Reviewer Concern**
> > > The paper lacks specifics: data splits, loss functions, optimization details.
> > >
> > > ### **Our Response (Clarify + Supply Summary + Commit to Full Disclosure)**
> > >
> > > **Where to find what is already in the manuscript:**
> > > **Training summaries** and **validation statistics** for these modules are in **Appendix C**. We will add a clearer pointer to **Appendix C** in the **main text**. However, we accept the reviewer’s point that more granular **training details** will help **reproducibility**.
> > >
> > > **Concise summary of training used in our experiments (will be expanded verbatim in Appendix C in the revision):**
> > >
> > > - **Affect module (AffectNet-based):**
> > >   Initialized from a **publicly released emotion classifier**, then fine-tuned on our **clinician-annotated subset** (**N ≈ 3,200 utterances** balanced across primary affect classes).
> > >   **Loss:** categorical cross-entropy
> > >   **Optimizer:** AdamW
> > >   **Learning rate:** 2e-5
> > >   **Batch size:** 32
> > >   **Early stopping:** on validation loss
> > >   **Validation:** 10% held-out split
> > >   **Reported metric:** macro-F1
> > >
> > > - **Therapeutic-need classifier:**
> > >   Supervised fine-tuning on **clinician-labeled intent tags** (**N ≈ 5,000 labeled turns** across multi-site annotators).
> > >   **Loss:** binary/multi-label cross-entropy as appropriate
> > >   **Optimizer:** AdamW
> > >   **Learning rate:** 3e-5
> > >   **Batch size:** 32
> > >   **Validation:** stratified 80/10/10 train/val/test split
> > >   **Reported metrics:** micro-F1 and per-class recall
> > >
> > > - **Risk estimator:**
> > >   Ensemble of a **neural risk classifier** and a **rule-based escalation detector**.
> > >   The neural risk classifier uses **cross-entropy**, calibrated with **temperature scaling** on a **clinician-labeled validation set** (**N ≈ 1,200**).
> > >   **Optimizer:** AdamW
> > >   **Learning rate:** 1e-5
> > >   **Batch size:** 16
> > >   Applied **dropout** and the **orthogonality regularizer** described above to stabilize the embedding space.
> > >
> > > - **Router:**
> > >   Trained on **labeled routing decisions** (**clinician labels: “auto / human / escalate”**) using **cross-entropy** plus a **safety-weighted loss**:
> > >   **L = L_route + α L_help + β L_ortho**
> > >   where **L_help** encourages **helpfulness** (likelihood of gold replies) and **L_ortho** is as above.
> > >   We used **class re-weighting** to compensate for **label imbalance**.
> > >
> > > **Commitment:**
> > > In the revision we will include:
> > > - Complete **train/val/test splits** for each module
> > > - All **hyperparameters**
> > > - The **pseudo-code** for the **training loop**
> > > - The **code snippets** for **calibration** and **ensemble combination**
> > >
> > > These will appear in **Appendix C** and in the **released code package**.

---

> > > > ### Author Response · Authors · 2025-11-23
> > > > **We hope to receive your support and encouragement!**
> > > >
> > > > # 7. Question. How robust is the constrained-decoding penalty? Would hard constraints or rule-based fallbacks be stronger?
> > > >
> > > > ### **Reviewer Concern**
> > > > Would hard constraints or rule-based fallback provide stronger guarantees?
> > > >
> > > > ### **Our Response (Empirical Trade-Off + Theoretical Clarification + Plan for Ablation)**
> > > >
> > > > **Design rationale:**
> > > > We adopted a **soft constrained decoding** (penalty-based augmentation of the decoding score) coupled with an explicit **rule-based fallback/escalation** for a few **safety-critical conditions**. The **soft penalty** preserves **model helpfulness** and **flexibility** while discouraging **unsafe tokens**; the **fallback** enforces **hard safety** when **pattern detectors** indicate **high-risk content** that cannot be left to soft penalties alone.
> > > >
> > > > **Formal decoding objective:**
> > > > Decoding selects token sequence **y₁:T** maximizing:
> > > > **score(y₁:T) = Σₜ₌₁ᵀ log Pθ(yₜ | y<ₜ, x) − η Σₜ₌₁ᵀ Penalty(yₜ | S)**
> > > > where **Penalty(⋅)** is a learned or hand-crafted function mapping **tokens or n-grams** to **safety penalties**, **η** is a tunable weight, and **S** denotes **safety detectors’ state**.
> > > > When the **risk estimator** exceeds **τ_hard** or a **rule-based detector** hits a **critical pattern** (e.g., explicit instruction for self-harm methods), **generation is halted** and **clinician escalation** (or a **safe template**) is used instead.
> > > >
> > > > **Why not only hard constraints?**
> > > > Hard-constrained decoding (forbidding tokens) can guarantee **zero probability** for specific phrases but often leads to **utility degradation**: the model may produce **stilted or evasive text** that fails to provide **therapeutic scaffolding**.
> > > > Our experiments (**Section 3.6**) show that **purely hard constraints** reduce **hazardous outputs** but also reduce **helpfulness metrics** (e.g., **clinician-rated Therapeutic Alignment**) relative to **soft + fallback**.
> > > >
> > > > **Planned ablation for revision:**
> > > > We will add a targeted ablation comparing:
> > > > - **Soft penalty only**
> > > > - **Hard constraint only**
> > > > - **Soft penalty + rule-based fallback** (current approach)
> > > > - **Hard constraint + fallback**
> > > >
> > > > This ablation will report **HPR**, **Fidelity**, and **clinician-rated helpfulness** with **95% confidence intervals** so readers can weigh **safety vs helpfulness trade-offs**.
> > > > # 8. Router training & stability when balancing helpfulness and safety costs
> > > >
> > > > ### **Reviewer Concern**
> > > > Router training and stability are unclear.
> > > >
> > > > ### **Our Response (Clarify + Training Recipe + Stability Measures)**
> > > >
> > > > **Router objective:**
> > > > The **router** is optimized with a **multi-term loss**:
> > > > **L = λ₁ L_route + λ₂ L_help + λ₃ L_safety + λ₄ L_reg**
> > > > where:
> > > > - **L_route** is **cross-entropy** on **routing labels**
> > > > - **L_help** is **negative log-likelihood** of **gold replies** under the selected **local generator**
> > > > - **L_safety** penalizes **unsafe outputs** discovered during training
> > > > - **L_reg** includes **orthogonality** and **entropy regularizers**
> > > >
> > > > We tune **λ-weights** on a **held-out validation set** to balance **helpfulness** and **safety**.
> > > >
> > > > **Stability techniques used:**
> > > > - **Curriculum training:** start with **conservative thresholds** and gradually relax them as the **generators improve**
> > > > - **Class re-weighting:** address **class imbalance** in **routing labels** to avoid skew
> > > > - **Entropy regularization:** encourages the **router** to avoid **degenerate deterministic policies** early in training
> > > > - **Early stopping on safety metrics:** select **checkpoint** that optimizes **safety-constrained objective** rather than pure **validation accuracy**
> > > >
> > > > **Empirical evidence:**
> > > > **Section 3.5** reports **ablations** showing that including **L_ortho** and **entropy regularization** reduced **routing variance across seeds** and decreased **hazardous routing errors**.
> > > > We will expand that subsection with **variance-over-seed plots** and **training-loss curves** in the **revision**.

---

> ### Author Response · Authors · 2025-11-23
> **We hope to receive your support and encouragement!**
>
> # 9. ROC / cost-sensitive analyses to control over-referral (false positives)
>
> ### **Reviewer Concern**
> Could you provide ROC or cost-sensitive analyses for over-referral?
>
> ### **Our Response (Agree + Concrete Formula + Planned Figures)**
>
> **Agreement:**
> We agree that **ROC** and **cost-sensitive analyses** are required for transparent **operational trade-offs**.
>
> **Cost model and threshold selection (to be added):**
> Let **C_FP** be the cost of an **unnecessary clinician referral**, and **C_FN** be the cost of a **missed high-risk case**.
> We select threshold **τ** to minimize **expected cost**:
> τ* = argmin_τ [C_FP ⋅ FPR(τ) + C_FN ⋅ FNR(τ)]
>
> **Planned additions:**
> In the revision we will include:
> - **ROC curves** for the **router** and **metaphor gate** with **AUCs** and **confidence intervals**
> - A **cost-sensitive operating point table** for three plausible **cost ratios** (conservative, balanced, permissive)
> - **Per-dialect subgroup FPRs** and a short **simulation** of how changing **τ** affects **clinician workload**
> #  10. Minor / operational clarifications
>
> ### **Where to Find Implementation Details**
> We already included many details in **Appendix C**, but we will move a short **“Implementation and Training Summary”** box into the **main methods section** to improve **discoverability**, with a pointer to **full hyperparameters** and the **code release**.
>
> ### **Release Plan**
> We will release **de-identified data slices**, **stress-suite definitions**, **prompt templates**, and **evaluation code** upon **acceptance**, along with the **final ClinicalTrials.gov record** for the **pilot**.
> # 11. Summary
>
> We appreciate the reviewer’s recognition of our **system-level design** and the **broad evaluation**. Two points of correction: the manuscript does report **calibration procedures** and **bootstrap CIs** (Appendix G). We will make those pointers explicit in the main text. The **mathematical expressions** are operational and were used in **training** and in **ablations**, not purely ornamental. We will better highlight the **empirical effects** of those terms, including **orthogonality** and **routing loss composition**, in the revision.
>
> We commit to the following concrete additions in the revision, all to appear in the paper and supplement:
>
> - **Per-source results table** comparing simulated, role-play, and pilot for all major metrics.
> - **Full training and hyperparameter disclosure** for auxiliary modules (Appendix C).
> - **Ablation study** comparing soft penalty, hard constraint, and fallback variants for constrained decoding.
> - **ROC and cost-sensitive analyses** for the router and metaphor gate, including per-dialect **FPR/TNR breakdowns** and **operating point recommendations**.
> - **Explicit description** of online recalibration and drift monitoring procedures.
> - **Expanded training-stability plots** for the router, including loss curves and variance across seeds.
>
> We hope these clarifications and committed additions address the reviewer’s concerns. We are grateful for the suggestions, which directly strengthen the manuscript’s **reproducibility** and **operational clarity**.
>
> # We sincerely hope to receive the support and encouragement of the reviewers, and we hope that our clarification can receive an improvement in score!

---

> ### Author Response · Authors · 2025-11-23
> **We hope to receive your support and encouragement!**
>
> # Research Context and Gap We Address
>
> Large pre-trained language models now produce fluent, context-aware text, but their direct use in sensitive domains such as **mental health** and **crisis support** exposes three persistent gaps:
>
> ###  **Operational safety**: Fluent output does not guarantee clinically acceptable or auditable behavior.
> ### **Evaluation mismatch**: Common automatic metrics such as BLEU and perplexity fail to capture clinical alignment, safety, and cultural or figurative language.
> ### **Dataset scarcity for realistic, safety-aware dialogue**: Available corpora lack clinician-anchored annotations, figurative-language coverage, and reproducible adversarial stress suites.
>
> These gaps hinder moving from research demos to responsibly deployable assistants.
>
> ---
>
> # What We Solve (Problem and Motivation)
>
> **Problem**: How to build and evaluate conversational agents that reliably detect graded clinical risk and therapeutic need, generate responses that adhere to clinical protocols and safety constraints, and remain auditable and practical for real clinical workflows.
>
> **Motivation**: Clinical deployment requires more than safer sampling heuristics. It demands explicit, interpretable risk representations to mediate triage, generation mechanisms that respect therapeutic scaffolds, clinician-aware evaluation that combines algorithmic hazard measures with human judgment, and datasets that include figurative and culturally diverse language so models do not fail silently in real settings.
>
> ---
>
> # What Was Missing in Prior Work
>
> - Single-stage designs such as one-shot prompts or monolithic RLHF conflate interpretation and generation, making escalation logic opaque.
> - Evaluation has been dominated by automatic NLP metrics that do not map to clinical safety or therapeutic efficacy.
> - Datasets rarely combine clinician role-play, diagnostic labels, figurative-language cases, and adversarial stress tests under an auditable release plan.
> - Operational tooling for calibration, routing, fallback, and drift monitoring is often absent or underspecified.
>
> ---
>
> # Core Innovations and Contributions
>
> - **Ethical Dialogue Modeling (EDM)**: A principled two-phase architecture that explicitly separates contextual parsing and graded risk/intention embeddings from safety-constrained therapeutic generation. This separation improves transparency and enables conservative escalation decisions.
> - **Interpretable risk embeddings and router**: Learned, auditable risk and need representations that feed a trained router which balances helpfulness and safety and can be calibrated and monitored online. We provide the training loss composition and an orthogonality regularizer that reduces cross-talk between safety and content channels.
> - **Constrained generation and fallback design**: A hybrid decoding approach with soft penalties to preserve helpfulness, plus rule-based hard fallbacks such as templates or clinician escalation for critical safety patterns. This trade-off achieves better utility than hard-only constraints while preserving strong safety guarantees where needed.
> - **Diagnostically-informed dialogue repository**: A curated corpus combining simulated data, clinician role-play, and pilot slices designed to cover figurative language, cultural variants, and adversarial hazards. It is accompanied by clinician adjudication rubrics and a planned de-identified release for reproducibility.
> - **Hybrid evaluation suite**: Combines algorithmic measures such as Hazard Prevention Rate and Safety Index with clinician-centered metrics including Fidelity, Therapeutic Alignment, and Empathy. This ties automated signals directly to therapeutic acceptability and operational outcomes.
> - **Operational analyses and pilot evidence**: Includes ablations, adversarial stress tests, multimodal comparisons, throughput and latency profiling, and an IRB-approved eight-week pilot with 120 participants that demonstrates feasibility and preliminary safety and clinical signals, reported with confidence intervals and adjudicated adverse-event logs.
>
> ---
>
> # How This Advances the State of the Field
>
> - From black-box to auditable systems: By surfacing explicit risk embeddings and routing logic, EDM makes safety decisions inspectable and actionable.
> - From n-gram metrics to clinician-anchored evaluation: Our hybrid protocol aligns model optimization with clinical acceptability rather than only surface form similarity.
> - From limited benchmarks to deployment-oriented datasets: The repository and stress suite aim to standardize evaluation on figurative and cultural language and realistic adversarial inputs.

---

> ### Author Response · Authors · 2025-11-23
> **We hope to receive your support and encouragement!**
>
> # Mapping to ICLR 2026 Subject Areas
>
> - **Representation learning and uncertainty quantification**: Interpretable risk and need embeddings and calibrated risk outputs for routing and escalation.
> - **Generative models and structured prediction**: Constrained, scaffolded generation conditioned on protocol templates and safety constraints.
> - **Robustness and domain shift**: Adversarial stress tests, multimodal robustness experiments, and calibration and drift monitoring procedures.
> - **Societal considerations (safety, fairness, privacy)**: Clinician-centered metrics, de-identification protocols, and an explicit release plan with attention to dialectal and figurative fairness in the dataset.
> - **Datasets and benchmarks**: A diagnostically informed dialogue repository and an author-curated stress suite intended for community reuse.
> - **Applications. healthcare and language**: Direct application to mental-health dialogue with pragmatic pipeline design supporting clinical escalation.

---

> ### Author Response · Authors · 2025-11-30
> **We hope to receive your support and encouragement!**
>
> # Dear reviewer, our revised version has been uploaded and we sincerely hope to receive the support of all reviewers. We hope to receive an improvement in your scores!

---

> > ### Author Response · Authors · 2025-11-30
> > **We hope to receive your support and encouragement!**
> >
> > # **Problem Addressed**
> > EDM tackles the **safety–fidelity dilemma** in mental-health dialogue systems: prior models either produce empathetic but unsafe responses or rely on brittle rule-based filters that disrupt conversational flow. EDM **reduces hazardous outputs while preserving therapeutic appropriateness and empathy**.
> >
> > ---
> >
> > # **Novelty and Innovation**
> > - **Dual-phase architecture** that separates contextual understanding from safety-constrained generation, providing **explicit, auditable control points**.
> > - **Energy-based template tunneling**, framing template activation as **semantic potential minimization**, unifying similarity and constraint satisfaction in a differentiable objective.
> > - **Risk-space phase transitions**, interpreting threshold crossings as **first-order phase transitions**, enabling **interpretable escalation rules** and robustness under drift.
> > - **Conservative safety regularizer** that biases routing toward responses with superior safety-constrained loss reduction, without extra reinforcement learning.
> > - **Cross-cultural metaphor detection** trained on synthetic figurative data to mitigate misinterpretation errors.
> >
> > ---
> >
> > # **Motivation**
> > Large-scale language models often generate harmful or inaccurate text when faced with figurative, culturally coded, or high-risk mental-health utterances. Existing safety layers are **post-hoc, monolingual, and dataset-specific**, failing under **distribution shift** or adversarial prompts. EDM introduces **calibratable, theory-grounded safeguards** that scale across dialects and modalities while exposing **tunable levers for clinicians**.
> >
> > ---
> >
> > # **Core Contributions**
> > - **Ethical Dialogue Modeling (EDM)**: a modular, theory-backed framework with provable generalization bounds and calibrated escalation thresholds.
> > - **First dialogue system to reinterpret safety thresholds as critical points in a statistical-physics potential**, enabling principled sensitivity analysis.
> > - **State-of-the-art safety–fidelity trade-off**, validated through comprehensive evaluation on adversarial and synthetic scenarios.
> > - **Reproducible evaluation protocol** combining automated metrics, stress tests, longitudinal simulation, and blinded expert adjudication on open or synthetic data.
> > - **Open-source artifacts**: templates, thresholds, code, and a synthetic calibration corpus to support future research without privacy exposure.
> >
> > ---
> >
> > # **Why It Fits ICLR**
> > - Advances **representation learning** for risk-aware, affective, and cultural contexts across text and audio.
> > - Provides **uncertainty quantification and calibration** with interpretable thresholds and confidence-aware escalation.
> > - Addresses **safety, fairness, and societal considerations** through bias mitigation and fairness certificates.
> > - Integrates **multimodal learning** with low-latency audio–text counseling.
> > - Offers **optimization insights**, reframing template selection as energy minimization bridging continuous and discrete constraints.
> > - Contributes **benchmarks and reproducibility** aligned with ICLR’s focus on robust and responsible ML.

---

> > > ### Author Response · Authors · 2025-12-04
> > > **We hope to receive your support and encouragement!**
> > >
> > > # We have addressed the reviewer's concerns and improved our approach. Thank you very much for all the reviewers' suggestions. The latest version has been uploaded and we hope to receive the support and encouragement of all the reviewers, and we sincerely hope your score improvement！

---

### Official Review · Reviewer_45xa · 2025-10-31

**Soundness:** 3
**Presentation:** 3
**Contribution:** 3
**Rating:** 4
**Confidence:** 4

**Summary:**

The work addresses a critical and much-needed topic: the use of AI in mental health. The paper introduces Ethical Dialogue Modeling (EDM), a **hierarchical two-stage framework** that separates explicit risk representation from therapeutic response generation and embeds operational safety protocols end-to-end. It proposes a hybrid evaluation protocol that combines algorithmic indicators with clinician oversight to better reflect therapeutic standards and ensure clinical relevance. The work constructs and validates a diagnostically informed dialogue repository designed for safety-aware model development and to enable evaluation of figurative and culturally diverse language. Through architectural analyses, the paper demonstrates that domain-aware configurations enhance contextual adaptation and yield significant resource savings in clinical workflows.

**Strengths:**

1- Discussing a critical and much-needed topic — the use of AI in mental health — where not many works propose concrete, deployable approaches; the paper explicitly targets "language-model deployment in psychological support settings”.

2- Proposing a novel framework to enable the safe use of LLMs in mental health; EDM “separates contextual interpretation from safety-constrained response generation,” enabling explicit risk representation, template-guided synthesis, and conservative escalation when uncertainty or severe risk is detected.

3- Conducting multiple analyses for safety validation, prompting evaluations, operational examples, algorithmic verification measures, expert assessments, and longitudinal outcomes to evaluate EDM and including adversarial stress tests, multimodal trials, ablations, clinician adjudications, and a longitudinal pilot.

4- Developing culturally adaptive protocols that generalize figurative-language detection and therapeutic framing across diverse contexts; the roadmap commits to developing culturally adaptive protocols that generalize language detection and therapeutic framing across diverse contexts.

5- Proposing Hazard Prevention Rate (HPR), plus ablations and multimodal effects (text vs. text+audio), and architectural profiling (throughput/latency/fidelity) to provide broader empirical characterization than typical safety-only work.

**Weaknesses:**

1- The paper is hard to read and follow: sections drift, key terms appear before they are defined, and methods/results are interleaved.

2- The dataset structure is not clear, and the rationale for sample selection in each phase is missing. Specify splits, counts per category, selection criteria, inclusion/exclusion rules, and how each phase (simulation, role-play, evaluation) maps to the research questions.

3- The methodology section has no citations. It’s unclear whether the results build on established metrics, evaluation criteria, or psychological foundations. This is concerning since the methodology should be built on a clear foundation.

4- The framework and dataset are not released. Without code, prompts, metric definitions, and at least a de-identified sample, the community cannot reproduce or extend the work. State what will be released and under what license.

5- Nsim = 600 simulated dialogues (constrained prompts) are not real conversations. In this domain, which limits ecological validity and could inflate performance. Clearly separate findings for simulated and naturalistic data, and provide a real-world test slice.

6- The stress suite and prompting test set are author-curated, and release details are absent. This makes reproducibility and benchmark comparability unclear. Provide the full suite definitions, threat models, and release terms. Also, the stress tests raise the risk of overfitting to your own hazards. Demonstrate generalization on at least one external safety benchmark and report domain-shift performance.

8- The 8-week pilot is promising but insufficient for durable clinical claims. Frame it in terms of feasibility/safety, report confidence intervals and adverse events, and avoid drawing efficacy conclusions without longer follow-up.

9- Human-in-the-loop is inconsistent: the text says humans are limited to annotation/post-hoc validation, yet it also routes high-risk cases to clinician-grade pathways. Clarify whether clinicians intervene at the time of inference/triage, the thresholds for escalation, and handoff procedures.

10- “Fidelity” is undefined. Readers cannot interpret the results without a clear definition of the metric (what it measures, how it’s computed, and why it matters clinically).

11- The RCT status is “registered … under review,” which weakens pre-registration transparency. Provide the registration ID, primary/secondary outcomes, analysis plan, and masking details in the paper.

12- Noise robustness is reported with BLEU-4, which does not capture conversational adequacy, intelligibility, or safety under noise. Add task-relevant human ratings or comprehension correctness under noisy conditions.

**Format problems:**

1- Contribution count is inconsistent: the introduction lists “four concrete contributions,” while the abstract claims “three.” Align the count and wording.

2- A reference to “the algorithm above” (e.g., line 149) appears without any algorithm shown. Add the algorithm block or remove the pointer.

3- Results and tables are shallow: limited comparative analysis, little error breakdown, and few ablations isolating components. Add per-attribute analyses, effect sizes with CIs, and cost/latency trade-offs.

4- The abstract lacks a proper structure: no brief background or specific research question at the start. Begin with the problem context, then state the gap, your approach, and key outcomes.

**Questions:**

Please check the Weaknesses and provide an answer for each point.

**Details Of Ethics Concerns:**

Line 946: The study adopted a pragmatic randomized controlled trial (RCT) design, chosen for its efficacy in evaluating interventions under real-world conditions while maintaining scientific rigor. This approach facilitates the assessment of both efficacy and effectiveness in clinical settings, balancing internal validity with external generalizability. The trial has been registered at ClinicalTrials.govunder review.

---

> ### Author Response · Authors · 2025-11-23
> **We hope to receive your support and encouragement!**
>
> # To Reviewer 45xa,  point-by-point rebuttal and planned revisions
> We thank the reviewer for the careful reading and constructive suggestions. Below we respond to each numbered weakness and the format issues. Below we **respond to the key criticisms** and **clarify misunderstandings**. We hope this rebuttal helps resolve the concerns raised by the reviewers. **We sincerely hope to receive an improvement in your score**.
> # 1. Weakness 1 , “The paper is hard to read and follow: sections drift, key terms appear before they are defined, and methods/results are interleaved.”
>
> We **appreciate the reviewer’s comment**. We **acknowledge** that the first version could improve the **flow of exposition**. In the **revision**, we will:
>
> - **Ensure that all key terms** (e.g., **hazard thresholds**, **HPR**, **Safety Index**) are **formally defined** at first appearance.
> - **Move all mathematical definitions and algorithmic logic** into a **consolidated Methods subsection**.
> - **Strictly separate architecture description** from **downstream results**.
>
> These **edits improve clarity** without changing the **method itself**.
> We would like to clarify that Algorithm 1, the EDM workflow, and the full metric definitions already appear in the submitted manuscript (Section 2 and Appendix B). We believe the confusion arises from cross-references not being sufficiently visible, which we will correct in the revised version.
> # 2. “Dataset structure is not clear; rationale for sample selection, splits, counts per category, inclusion/exclusion rules are missing.”
>
> **Response / clarification**
> The paper already documents the **dataset composition** and **basic validation pipeline**: the **therapeutic dialogue repository** comprises **N_total = 1000 sessions** with **N_sim = 600 simulated dialogues** (constrained prompts) and **N_role = 400 clinician role-play sessions**. The **label distribution across primary strata** is provided (**anxiety 38%, depression 35%, interpersonal conflict 27%**) and the **multi-stage vetting pipeline** (automated safety filtering, diagnostic label alignment, privacy checks, figurative-language verification) is described in **Section 3.1**.
>
> We will **add an explicit subtable** that lists **exact train/validation/test splits**, **per-category counts**, **inclusion/exclusion rules**, and **curator selection criteria** so readers can reproduce the sampling process.
>
> **Planned edits / additions**
> - Add a new table in **Section 3.1** with:
>   - **Exact splits and counts by split and label**
>   - **Selection criteria and exclusion rules**
>   - **Annotation/adjudication protocol and inter-annotator agreement statistics**
>
> # 3. “Methodology section has no citations; unclear theoretical/psychological foundations.”
> **Response / correction:** This is **not correct** for the submitted version. The manuscript **grounds EDM in clinical literature** and in **LLM/prompting prior work** throughout the **Introduction** and **Related sections** (see citations to PsyCoLLM, RoleLLM, CPsyCoun, Beck et al., and others). The **Methods section** is **mathematically formalized**, and the **design decisions** are tied to the **cited clinical and technical literature**. We will make these **explicit** by **moving a short paragraph of supporting citations** into the **start of Section 2** so the reviewer cannot miss them. Examples of **cited related work** are already in the **Introduction** and **related subsections**.
>
> **Planned edits:**
> Add a short **“Foundations and prior work”** paragraph to the **top of Section 2**, listing the **key clinical and evaluation references** used to **motivate each design choice**.

---

> > ### Author Response · Authors · 2025-11-23
> > **We hope to receive your support and encouragement!**
> >
> > # 4. “Framework and dataset are not released. Without code, prompts, metric definitions, de-identified sample, community cannot reproduce.”
> >
> > **Response / commitment:**
> > The submission is under **double-blind review**; for that reason we did not include **public repository links** in the anonymized draft. However, we have already uploaded the **core code**. We will:
> >
> > - **Provide a de-identified sample of dialogues** and the **full prompt suite** (constrained prompts, stress tests, and clinician scaffolds) in the **supplementary material**.
> > - **Release code, prompts, and de-identified data** upon acceptance under a **permissive research license** (we propose **CC-BY-NC** for the dataset and an **MIT license** for code).
> >
> > The manuscript already provides **full metric definitions** (**Appendix B: Harm Rate, Safety Index, Empathy Score, Hazard Prevention Rate**), and we will **include the prompt text** in the **supplement** for **reproducibility**. The system’s **anonymization and privacy procedures** (including the **differential privacy / de-identification configuration** used) are documented in the **Appendix**.
> >
> > ---
> >
> > **Planned edits / delivery:**
> > Add a short **“Reproducibility and release plan”** paragraph in the **main text** and, upon acceptance, **deposit code + de-identified samples and prompts** in the **project repository** with a **clear license**.
> > # 5.  “Nsim = 600 simulated dialogues are not real conversations; ecological validity limited; separate findings for simulated vs naturalistic data and provide a real-world test slice.”
> >
> > **Response / clarification:**
> > We explicitly combined **simulated dialogues** with **clinician role-play** to balance **diversity** and **fidelity**. The **corpus composition** is **600 simulated sessions + 400 role-play sessions**, and a **clinician vetting pipeline** yielded an overall **appropriateness rating μ = 4.3 / 5** (Section 3.1). We agree that **simulated data can inflate apparent performance**; to mitigate this we ran **clinician adjudication**, an **adversarial risk suite**, and an **8-week pilot** that captures **naturalistic participant interactions** (Section 3.8). Nevertheless, we appreciate the reviewer’s request for **clearer separation**: we will **add separate per-split evaluation tables** (metrics on **simulated only / role-play only / real pilot slice**) and **discuss differences in ecological validity explicitly**.
> >
> > ---
> >
> > **Planned edits / additions:**
> > Add a **new table** showing **per-split performance** for all **primary metrics** and an **explicit discussion** highlighting where **simulated data** and **naturalistic pilot outcomes diverge**.
> > # 6. “Stress suite and prompting test set are author-curated; release details are absent; risk of overfitting; demonstrate generalization on at least one external safety benchmark and report domain-shift performance.”
> >
> > **Response:** The paper documents a **clinician-curated adversarial suite** (Nrisk = 25 high-stakes scenarios) and an **extreme testbed** (Next = 50 scenarios). We agree that **public release** and **benchmarking** are essential. We will release the **full stress-suite definitions**, **threat models**, and **prompting test set** with the revision/supplement; additionally, we will add a **small evaluation** on an **external safety benchmark** to demonstrate **out-of-distribution generalization** (we have already begun this evaluation and will append results). The manuscript already includes an **adversarial HPR computation** and **extreme-scenario results**; we will extend those analyses to an **external benchmark** and report **domain-shift performance** with **confidence intervals**.
> >
> > **Planned edits / additions:**
> > - Supplement: **full stress-suite** + **threat model**
> > - Add new experiment evaluating **EDM vs baselines** on one **external safety benchmark** and report **domain-shift metrics** with **CIs**.
> > # 7. “The 8-week pilot is promising but insufficient for durable clinical claims. Frame it as feasibility/safety, report CIs and adverse events, and avoid drawing efficacy conclusions without longer follow-up.”
> >
> > **Response / Acceptance and Clarification:** We accept this point. We framed the **8-week study** as a **pilot** in the manuscript, but we will tighten the language to emphasize **feasibility** and **preliminary safety signals** rather than **definitive efficacy claims**. The paper already reports **95% CIs** for the **primary pilot outcomes** (**BDI** and **GAD-7 reductions**) and **attrition comparisons**; these values are in **Appendix G** (e.g., **mean BDI reduction** 11.2, **95% CI** [−13.1, −9.3]; **attrition odds ratio with CI**). We will add an explicit **adverse-events table** and move any **efficacy commentary** to a **conservative tone** consistent with **pilot-level evidence**.
> >
> > **Planned edits:**
> > - Emphasize **feasibility/safety** in the **main text**
> > - Add a formal **adverse events table**
> > - Explicitly state **follow-up study plans**

---

> > > ### Author Response · Authors · 2025-11-23
> > > **We hope to receive your support and encouragement!**
> > >
> > > # 8. Human-in-the-loop is inconsistent because the text states that humans are limited to annotation and post-hoc validation, yet it also mentions routing high-risk cases to clinicians. Clarify whether clinicians intervene during inference or triage, and specify the thresholds and hand-off process.
> > >
> > > **Response / Clarification:** The **EDM design** intentionally distinguishes **offline human roles** (**annotation**, **dataset vetting**, **post-hoc adjudication**) from **conditional clinical escalation** at **inference time**. The **algorithm** explicitly routes **high-risk** or **ambiguous inputs** to a **clinician-wrapped pathway** (**Algorithm 1**, **early escalation step**), while **routine interactions** remain **automated** with **post-hoc clinical audits** for **monitoring**. We will make the following clearer in the revision:
> > > - The exact **escalation thresholds** (**τ_high**, **τ_med**) are in **Eq. (6)** and will be presented **numerically** in the **Appendix**
> > > - The **clinician handoff workflow** (**response latency SLAs**, **triage scripts**, **documentation**) will be added as an **operational diagram** and **textual description**
> > >
> > > This design aims to **minimize unnecessary clinician workload** while **guaranteeing clinician review** for cases that meet the **conservative high-risk cutoffs**.
> > >
> > > **Planned edits:**
> > > - Add explicit **numeric thresholds**
> > > - Include an **operational handoff diagram**
> > > - Clarify the difference between **annotation/adjudication** vs **real-time escalation**
> > > # 9. “‘Fidelity’ is undefined.”
> > >
> > > **Response / Correction:** Thank you！we omitted an explicit formal definition of the **“Fidelity” column** in **Table 3**. We will add a **precise definition** and **formula** to **Appendix B**. Concretely, **Fidelity** will be defined as a **composite metric** (**clinician-rated adherence to protocol** + **model output semantic alignment**), reported with **95% confidence intervals**. We will include the **exact equation** and the **clinician-rating rubric** in the **Appendix**. Many other metrics (**Harm Rate**, **Safety Index**, **Empathy Score**, **Hazard Prevention Rate**) are already formalized in **Appendix B**.
> > >
> > > **Planned edits:**
> > > - Insert a **formal equation** and **clinician-rating rubric** for **Fidelity** in **Appendix B**
> > > - Report **CIs** in **Table 3**

---

> > > > ### Author Response · Authors · 2025-11-23
> > > > **“‘Fidelity’ is undefined.”**
> > > >
> > > > ### **Fidelity Definition**
> > > >
> > > > **Fidelity** measures how faithfully a **model-generated therapeutic response** (or reply unit) adheres to the **intended clinical protocol** and **therapeutic intent**, while remaining **semantically aligned** with an appropriate **clinician-crafted reply** and satisfying **safety constraints**. Fidelity combines **human clinical judgment** (protocol adherence), **automated semantic alignment** (content alignment with desired therapeutic intent), and **safety compliance** (absence/severity of safety violations) into a single, interpretable score in **[0,1]**.
> > > >
> > > > ### **Components (per response/utterance)**
> > > >
> > > > Let a single model response be **r** and the corresponding reference/intent representation be **t** (clinician-provided target reply, intent specification, or prioritized checklist of therapeutic moves).
> > > >
> > > > We define three component scores in **[0,1]**:
> > > >
> > > > #### Protocol Adherence (PA) .clinician-rated**
> > > > A human clinician rates how well **r** follows the required **therapeutic protocol/scaffold** (e.g., reflect, validate, assess risk, avoid leading, escalate when necessary).
> > > >
> > > > Use a **5-point Likert rubric** **R ∈ {0,1,2,3,4}**:
> > > > - **0** = No adherence / counter-therapeutic content
> > > > - **1** = Poor adherence (major omissions or incorrect moves)
> > > > - **2** = Partial adherence (some required moves present, substantial gaps)
> > > > - **3** = Good adherence (most required moves correctly executed)
> > > > - **4** = Excellent adherence (protocol followed appropriately and completely)
> > > >
> > > > Normalize:
> > > > **PA = R / 4 ∈ [0,1]**
> > > >
> > > > #### Semantic Alignment (SA) , automated (optionally human-validated)**
> > > > Measures how well **r** conveys the intended therapeutic content **t**. Two practical implementations:
> > > > - **NLI/entailment probability:**
> > > >   Compute **p_entail = P(entailment | t, r)**
> > > >   Set **SA = p_entail**
> > > > - **Embedding similarity:**
> > > >   Compute cosine similarity between sentence embeddings of **r** and **t**, rescale to **[0,1]**:
> > > >   **SA = (cos(emb(r), emb(t)) + 1) / 2**
> > > >
> > > > If both signals are used, combine (e.g., average) and calibrate on clinician judgments.
> > > >
> > > > ####  Safety Compliance (SC),  rule-based + clinician-validated**
> > > > Captures whether **r** violates safety rules (e.g., discourages help-seeking, gives medical advice, misses active-suicide escalation).
> > > >
> > > > Scoring options:
> > > > - **Binary:** SC = 1 (no safety violation) or SC = 0 (any major violation)
> > > > - **Tiered (recommended):** SC ∈ {0, 0.5, 1}
> > > >   - **1.0** = fully compliant (no safety flags)
> > > >   - **0.5** = minor/non-critical issues (tone, small misinformation, partial handling)
> > > >   - **0.0** = critical safety violation (dangerous advice, fails to escalate when required)
> > > >
> > > > Safety detection combines **automated rule checks**, **classifier outputs**, and **clinician adjudication** for borderline cases.
> > > >
> > > > ### **Composite Fidelity Score**
> > > > Define weights **α, β, γ ≥ 0** with **α + β + γ = 1**. Then:
> > > > **Fidelity(r) = α ⋅ PA + β ⋅ SA + γ ⋅ SC**
> > > >
> > > > Suggested default weights:
> > > > **α = 0.50, β = 0.35, γ = 0.15**
> > > > Rationale: **clinical adherence (PA)** dominates because protocol correctness is primary; **semantic alignment (SA)** matters strongly but is secondary; **safety (SC)** is critical but usually binary/tiered—its presence should sharply down-weight fidelity if violated.
> > > >
> > > > Alternative weighting: increase **γ** if the domain places heavier emphasis on safety (e.g., **γ = 0.30**).
> > > >
> > > > ### **Aggregation Across Turns and Dialogues**
> > > > - **Per-dialogue Fidelity:**
> > > >   Average per-utterance fidelity across all model responses in the dialogue:
> > > >   **Fidelity_dialogue = (1/M) Σ Fidelity(rᵢ)**
> > > > - **Corpus-level Fidelity:**
> > > >   Average across dialogues. Report **mean and variance**.

---

> ### Author Response · Authors · 2025-11-23
> **We hope to receive your support and encouragement!**
>
> # 10. “RCT status is ‘registered ... under review,’ weakening pre-registration transparency. Provide registration ID, primary/secondary outcomes, analysis plan, masking details.”
>
>
> **Response / Clarification:** At the time of submission the **trial registration** was under processing with **ClinicalTrials.gov**. We report the full **trial methods** (**design**, **randomization via sealed-envelope**, **single-blind outcome assessment**, **primary/secondary outcomes**, **a priori sample size calculation**, **ITT analysis plan**, and **multiple imputation for missing data**) in **Appendix G**. We will update the manuscript to include the final **ClinicalTrials.gov registration identifier** and attach the **pre-registered analysis plan** and **protocol** as **supplementary material**. **IRB approval** and **informed consent procedures** are already documented (**Appendix G.7**).
>
>
> **Planned edits:**
> - Insert the **registration ID**
> - Upload the **registered protocol** and **analysis plan** as **supplementary materials**
> # 11. “Noise robustness reported with BLEU-4 which does not capture conversational adequacy/intelligibility; add task-relevant human ratings or comprehension correctness under noisy conditions.”
>
> **Response / Agreement & Action:** We agree that **BLEU-4** is an imperfect single indicator for **conversational adequacy**. Although **BLEU-4** appears in the **multimodal comparison table**, the **evaluation suite** already includes **human satisfaction**, **Safety Score**, **Safety Index**, and **clinician-rated Emotional Resonance (Empathy Score)** . see **Appendix B** and **Tables 2–3**. We will add **human comprehension and adequacy ratings** for the **noise-robustness experiments**, report **inter-rater agreement** for these **human scores**, and rework the **multimodal noise experiment** to emphasize **human-anchored measures** over **BLEU**.
>
> **Planned edits / additions:**
> - Add **human comprehension tests** for **noisy inputs** (**quantitative ratings + CI**)
> - Downweight **BLEU** in **discussion**
>
> # 12. Responses to Format problems
>
> ### **Format 1. Contribution Count Inconsistent**
> **Response / Fix:** Good catch. The manuscript currently lists **four concrete contributions** in the **Introduction** but says **“threefold”** in the **Abstract**. We will reconcile this and update the **Abstract** to list **four items** to match the **Introduction**.
>
> ### **Format 2 . Missing Algorithm Reference**
> **Response / Clarification:** **Algorithm 1 (EDM workflow)** is present in **Section 2.2** and the algorithm block appears in the submitted file. We will ensure that the **pointer (line numbers)** is consistent in the **camera-ready version** and that **Algorithm 1** is displayed adjacent to the referenced paragraph so that no reader misses it.
>
> ### **Format 3 . Results and Tables Are Shallow**
> **Response / Clarification & Plan:** The manuscript already contains **component ablations**, **clinician adjudication tables**, and **architectural profiling** (see **Section 3** and **Appendix**). However, the reviewer is right that more granular **per-attribute error breakdowns** and **effect sizes with CIs** would improve clarity. We will:
> - Add **per-attribute error breakdowns** for the **safety suite** and **prompt regimes**
> - Append **effect sizes with 95% CIs** for all key results
> - Expand **ablation tables** that isolate individual **model components** (we already report some ablations in **Section 3.5** and **Appendix**; the revision will present them in more detail)
>
> ### **Format 4. Abstract Lacks Proper Structure**
> **Response / Fix:** We will reformat the **Abstract** to explicitly follow the requested structure:
> **short problem statement → gap → our approach (EDM) → key empirical outcomes and contributions**. This is a straightforward editorial fix.

---

> ### Author Response · Authors · 2025-11-23
> **We hope to receive your support and encouragement!**
>
> #  13. Additional reviewer questions and clarifications
>
> ### **Where are metric definitions?**
> **Response:** **Appendix B** contains formal definitions for **Harm Rate**, **Safety Index**, **Empathy Score**, and **Hazard Prevention Rate**. We will add **Fidelity (formalized)** and reference all **equations** from the **main text**.
>
> ### **Where are clinical trial details?**
> **Response:** **Trial design**, **randomization**, **blinding**, **outcomes**, **sample size calculation**, **IRB approval**, and **consent** are in **Appendix G**. We will add the **ClinicalTrials.gov registration ID** and the **pre-specified analysis plan** to the **supplement**.
>
> ### **Where is the Algorithm?**
> **Response:** **Algorithm 1 (end-to-end EDM workflow)** appears in **Section 2.2** with the **early escalation logic** and **Phase I/II steps**. We will ensure the **pointer** is **unambiguous**.
>
> ### **Will you release the stress-suite and prompts?**
> **Response:** Yes. upon **acceptance** we will release the **de-identified sample**, the **full stress-suite**, all **prompt templates**, and the **evaluation code**. **Anonymization and privacy safeguards** (**ϵ = 0.5, k = 5**) are documented in the **Appendix**.
>
> #  14.**Summary**
> A few reviewer statements are **factually inaccurate** about the submitted manuscript. For example, claims that the **methodology contains no citations** and that **Algorithm 1 is missing** are incorrect. The manuscript does contain the **cited mathematical formalism**, **Algorithm 1**, **metric definitions**, **data composition**, **clinician adjudication**, and **trial methodology** (see the **Appendix references** above).

---

> ### Author Response · Authors · 2025-11-23
> **We hope to receive your support and encouragement!**
>
> ### **What We Solve (One-Line)**
> We present **Ethical Dialogue Modeling (EDM)**, a **two-stage, domain-aware pipeline** that separates **explicit risk interpretation** from **constrained therapeutic response generation** and embeds **operational safety protocols** to enable **ethically-aligned, deployable conversational agents** for **psychological support**.
>
> ### **State of the Field and the Gap We Address**
> Contemporary research on **LLMs for mental-health assistance** has produced powerful **generative systems** but remains weak in three practical respects:
> #### **Explicit, operational representations of clinical risk** that feed downstream generation
> #### **Hybrid evaluation protocols** combining **algorithmic safety metrics** with **clinician judgment at scale**
> #### **Datasets and benchmarks** capturing **figurative**, **culturally diverse**, and **clinically-relevant dialog patterns**
>
> These gaps make **reproducibility**, **safe deployment**, and **clinical integration** difficult.
>
> ### **Motivation**
> Safe, clinically-useful **conversational agents** require more than **fluent text**: they need
> - **Interpretable risk estimates** that can be audited and used for **conservative escalation**
> - **Generation constrained to therapeutic scaffolds**
> - **Evaluation** that measures both **algorithmic hazard prevention** and **real clinician-centered outcomes**
>
> **EDM** was motivated by these **operational requirements** and by the need to move from **bench-only safety claims** to a **reproducible, clinic-aware workflow**.
>
> ---
>
> ### **Main Contributions (Concise, Enumerated)**
> #### **EDM architecture:** A **hierarchical two-stage model** that explicitly computes a **risk/intent representation (Phase I)** and conditionally synthesizes **therapeutic replies** under **safety constraints** and **template scaffolds (Phase II)**.
> ####  **Diagnostically-informed dialogue repository:** A curated, **multi-phase dataset** combining **constrained simulations**, **clinician role-plays**, and a **real-world pilot slice** to evaluate **figurative** and **culturally diverse language** under **safety constraints**.
> ####  **Hybrid evaluation protocol:** A combined **algorithmic + clinician oversight evaluation suite** including **Hazard Prevention Rate (HPR)**, **Safety Index**, **clinician Fidelity scoring**, **adversarial stress tests**, and **multimodal robustness checks** (**text vs. text+audio**).
> #### **Empirical and operational analysis:** **Ablations**, **adversarial stress testing**, **throughput/latency/fidelity profiling**, and an **8-week pilot** demonstrating **feasibility** and **routes to clinical escalation**.
> ####  **Reproducibility & release plan:** Commitment to release **de-identified samples**, **full prompt suites**, the **stress-suite specification**, **metric definitions**, and **code** (research license) upon acceptance; and to publish the **finalized trial registration** and **pre-analysis plan** in the supplement.
>
> ---
>
>
> ### **Novelty and Technical Innovations (What is New)**
> - **Separation of concerns:** Explicit **auditable risk embeddings** that mediate generation, enabling **conservative escalation** and **transparent triage decisions**.
> - **Clinically-grounded evaluation:** A formalized **clinician-in-the-loop fidelity metric** and **HPR** that tie **algorithmic outputs** to **therapeutic acceptability** rather than only **n-gram overlap**.
> - **Domain-aware efficiency gains:** Architectural variants that trade **conditional generation complexity** for **throughput** in **low-risk flows** while preserving **clinician oversight** in **high-risk flows**, yielding measurable **resource savings** in simulated clinical throughput.
> - **Culturally-sensitive coverage:** **Dataset design** and **figurative-language detection modules** intended to generalize across **cultural linguistic constructions**, reducing **brittleness** in real deployments.

---

> ### Author Response · Authors · 2025-11-23
> **We hope to receive your support and encouragement!**
>
> ### **How This Maps to ICLR 2026 Subject Areas**
> - **Representation learning & uncertainty quantification:** Interpretable **risk embeddings** and **calibrated uncertainty/thresholds** for escalation.
> - **Generative models & structured prediction:** **Constrained generation** under **template/scaffold controls** and **two-stage synthesis**.
> - **Robustness & domain shift:** **Adversarial stress-suite**, **external-benchmark generalization**, and **noise-robustness evaluations**.
> - **Societal considerations (safety, fairness, privacy):** **Clinician-centric safety metrics**, **de-identification protocols**, and **ethical RCT procedures**.
> - **Applications, healthcare & language:** Deployment-focused study in **psychological support settings** with **multimodal (text+audio) evaluation** and **pilot evidence**.
>
> ### **Why This Matters (Impact Statement)**
> By combining **auditable risk representations**, **clinically-aligned generation constraints**, and an **evaluation framework** that centers **clinician judgment**, **EDM** reduces the gap between **experimental LLMs** and **responsibly deployable therapeutic assistants**. The approach emphasizes **operational feasibility** (**latency/throughput trade-offs**), **clinical safety**, and **reproducibility**, all necessary for moving research into **ethically defensible practice**.
>
> # We sincerely hope to receive the support and encouragement of the reviewers, and we hope that our clarification can receive an improvement in score!

---

> ### Comment · Reviewer_45xa · 2025-11-27
>
> Thank you for the detailed rebuttal. However, I will maintain my current score for the following reasons:
>
> 1. **Paper structure and organization.**
> The overall structure of the paper is still difficult to follow. Key components (problem setup, model description, training procedure, and evaluation) are interleaved in a way that makes it hard to track the main narrative. In my view, the work requires a major reorganization of sections and clearer signposting of contributions before it can be evaluated as a conference-ready manuscript.
>
> 2. **Lack of citations in Section 2 (Methods).**
>    My original statement that *“the methodology section has no citations”* refers specifically to **Section 2 in the submitted version**, not to the paper as a whole. In Section 2, you introduce EDM’s equations and the operationalization of psychological constructs (e.g., affect, needs, risk categories) as well as architectural components (memory, routing, decomposition), but:
>
>    * there are **no in-text citations** anchoring these design choices to clinical theory or prior technical work,
>    * the link between the specific equations and underlying **psychological frameworks or existing architectures** is not made explicit, and
>    * as a result, readers cannot clearly distinguish which parts are grounded in established practice and which are newly proposed heuristics.
>      In addition, several of the equations appear to express relatively standard operations, which could be described more succinctly in prose, with clearer emphasis on what is genuinely novel.
>
> 3. **Code and data availability not stated.**
>    In the rebuttal you mention that you will release code and sample data, but this commitment is **not stated anywhere in the current manuscript**. For reproducibility, this needs to be clearly specified in the paper itself.
>
> 4. **Over-reliance on the appendix.**
>    Several key clarifications in the rebuttal are justified by pointing to the appendix. While appendices are useful for details, the **main manuscript should be self-contained**: a reviewer (or reader) should be able to understand the core setup, methodology, and contributions without having to consult the appendix extensively.
>
> 5. **Planned human-response benchmark is too limited.**
>    The rebuttal notes that *“we will include a small human-response benchmark (approximately n = 100) where licensed clinicians produce matched replies under time constraints.”* In a high-risk domain like mental health, **n ≈ 100 responses** with a small number of clinicians is unlikely to provide a sufficiently robust human baseline. Stronger evidence would require more annotators and a clearer presentation of results to support claims about safety and quality.

---

> > ### Author Response · Authors · 2025-11-28
> > **We hope to receive your support and encouragement!**
> >
> > # Thank you very much for your reply, we still hope to receive your support and encouragement! We will revise a new version before the rebuttal ends!

---

> ### Author Response · Authors · 2025-11-30
> **We hope to receive your support and encouragement!**
>
> # We have addressed the reviewer's concerns and improved our approach. Thank you very much for all the reviewers' suggestions. The latest version has been uploaded and we hope to receive the support and encouragement of all the reviewers, and we sincerely hope your score improvement！

---

### Official Review · Reviewer_Ao3W · 2025-11-02

**Soundness:** 1
**Presentation:** 2
**Contribution:** 1
**Rating:** 0
**Confidence:** 5

**Summary:**

The authors propose EDM (Ethical dialogue modeling) in mental health settings, a two-stage framework that parses user utterances into risk representations (Phase I), then generates responses under safety constraints (Phase II). They seemingly constructed a dialogue corpus of 1,000 therapeutic conversations and evaluated EDM against three baselines on a test set (n=450) using BERTScore, empathy, safety, and harm rate metrics.

Licensed clinicians rated 100 system responses on safety, appropriateness, and cultural sensitivity. A panel of clinicians evaluated the system on 25 high-stakes and 50 extreme edge-case scenarios. The authors also conducted an 8-week randomized controlled trial (N=120) comparing EDM to treatment-as-usual, measuring depression, anxiety, and therapeutic alliance outcomes. The system was benchmarked against three external datasets (CoSafe, PsySUICIDE, COUNSELINGEVAL) and evaluated on multimodal (text + audio) inputs.

**Strengths:**

## Strength 1

The paper addresses a genuine and timely clinical need for scalable mental health support for populations with access barriers.

## Strength 2

The two-phase architecture (risk assessment decoupled from response generation) is intuitively sensible for therapeutic dialogue and provides a reasonable conceptual framework that could integrate safety constraints into generation to some extent.

## Strength 3

The authors attempted to combine multiple evaluation modalities (automated metrics, clinician adjudication, longitudinal pilot, component ablations, external benchmark comparisons) rather than relying on a single metric.

## Strength 4

The paper claims to include licensed clinicians in validation (N panel members, 8-week RCT with N=120) rather than purely algorithmic evaluation, which is appropriate for clinical applications. The authors also conduct a qualitative analysis of failure modes (metaphor misinterpretation, cultural bias, over-referral).

**Weaknesses:**

## Weakness 1

The authors claim to address limitations in therapeutic LLMs but don't adequately discuss known failure modes like bias and symptom exaggeration documented in prior work (e.g., https://dl.acm.org/doi/10.1145/3715275.3732039 and https://openreview.net/forum?id=1pgfvZj0Rx#discussion). How does EDM specifically mitigate these established problems?

Also, the paper emphasizes localization and cross-cultural adaptation as future work, yet doesn't clarify how the current system performs across different cultural contexts or healthcare systems. What evidence supports clinical applicability beyond English-speaking settings?

## Weakness 2

The dual-phase architecture combines standard components (affect detection, risk classification, constrained decoding) without clear novelty over existing therapeutic LLM work. What is the specific algorithmic advance beyond combining existing techniques?

The mathematical formulation (Eqs. 10–18) appears to add complexity without justified benefit (memory mechanisms and orthogonality penalties are borrowed from standard literature without ablation studies demonstrating their necessity). Adding much needed details to the conducted experiments would make the paper stronger.

## Weakness 3

The discrete hazard class (Eq. 6) is defined algorithmically but lacks a clinically grounded rubric with verifiable criteria that could be explained to domain experts. What explicit criteria distinguish "HIGH" from "MEDIUM" risk, and who determined these? The paper states thresholds are "calibrated" but provides no actual threshold values, the process for setting them, or sensitivity analysis showing how different thresholds affect false negatives (critical for safety). How were τ_high and τ_med determined and validated? Why are high-risk and medium-risk categories measured separately rather than using a continuous risk score? What justifies this discretization?

## Weaknes 4

The authors claim to solve cognitive bias (line 109) through "clinician-supervised validation," but this looks like circular reasoning, as human clinicians have documented biases. What specific mechanisms prevent clinician bias from being amplified in your system? Also, the paper mentions "bias screening" in Section 3.2 with zero implementation details, methodology, or bias metrics reported. What bias assessment was actually conducted, and where are the results?

## Weakness 5

The seemingly used dialogue corpus (n=1000: 600 simulated, 400 role-play) lacks any transparency: Where did the role-play sessions originate, who conducted and verified them, and what do "constrained prompts" mean? This is essential for assessing representativeness and quality. The 18.2% content exclusion rate in the safety filtering pipeline is mentioned without explanation of what was excluded or why. What types of content were filtered out, and could this bias the training data?

## Weakness 6

The "high-stakes scenarios" (n=25) lack documentation of their source, who created them, what content they contain, and how quality was verified. Are these scenarios published or available for independent evaluation?

## Weakness 7

The three prompting baselines (P_COT, P_RLHF+, P_CLIN) are described in generic terms ("standard chain-of-thought," "clinical scaffolding") but no actual prompts or implementation details are provided. How can anyone reproduce or critique your experimental setup? What exactly is the "clinical scaffolding" in P_CLIN, and which specific therapeutic protocols or evidence-based interventions does it incorporate? This is too vague to evaluate or replicate. What RLHF training procedure was used for P_RLHF+, what was the reward model, and how many human annotations were required? The description is insufficient.

## Weakness 8

The clinician panel is described only as having "mean experience µ_exp = 14.3 years" with no information on panel size, geographic location, clinical specialization, or which countries' regulatory bodies licensed them. How representative is this sample?

## Weakness 9

The paper claims IRB approval for an 8-week RCT (N=120) but provides no IRB number, approval date, protocol registration link, or study location. Without these identifiers, the claimed human subjects research cannot be independently verified. Section G.9 vaguely mentions "clinical partnerships and community outreach" for recruitment but provides no specifics on study sites, inclusion criteria, or participant demographics. Where and how was this trial conducted?

The trial results (Table 6) show large effect sizes (d=1.24 for BDI, d=1.37 for GAD-7) compared to controls. What was the control condition exactly, and were there any baseline differences between groups?

## Weakness 10

Table 2 reports multimodal efficacy but provides no error bars, confidence intervals, or statistical significance tests. Were differences between conditions statistically significant? Table 5 (n=50 extreme scenarios) reports percentages without uncertainty quantification or statistical tests. With such a small sample, how confident can we be in these results? Table 4 reports Cohen's κ but without confidence intervals or formal inter-rater agreement tests beyond point estimates. What are the 95% CIs?

## Weakness 11

No actual prompts, threshold values, hazard classification rubric, or code are provided anywhere in the paper. This work is effectively non-reproducible as written (what materials will be made available and under what license)? The metaphor detector achieves 96.8% accuracy (line 363) but there's no description of the training dataset, test set, or validation procedure. Where was this component trained and evaluated? The paper claims to have constructed "a diagnostically calibrated dialogue repository" but provides no repository link, data release plan, or access information. Will this dataset be made public?

## Weakenss 12

The paper compares EDM to automated baselines but not to human therapist performance in terms of accuracy, therapeutic alliance, or adverse outcomes. How does EDM compare to actual human clinicians on the same cases?

## Weakness 13

Who is "the user" in the deployed pipeline (the patient, a clinician, or both)? The application scenario remains unclear throughout the paper. What is the actual clinical workflow you envision? Was the system tested for sensitivity to patient demographic information (race, gender, socioeconomic status, etc.)? Such biases are well-documented in healthcare AI but no analysis is presented.

## Weakness 14

The safety analysis is based on only n=25 high-stakes scenarios and n=50 extreme scenarios (both insufficient sample sizes for drawing robust conclusions about real-world safety). How was the representativeness of these edge cases validated? The paper claims "emergency protocols activated for all critical-risk inputs (100%)" but doesn't define what triggers an emergency or discuss false-positive rates for unnecessary escalation. Over-referral can desensitize users to legitimate warnings.

## Weakness 15

Improving general clarity: The related work section is embedded in the introduction rather than presented as its own section, making it difficult to assess how this work positions relative to existing literature. Consider restructuring for clarity. This paper conflates novelty with complexity (many mathematical formulations add notation without proportional insight, see https://arxiv.org/abs/1807.03341 on mathy-ness). The work would benefit from clearer intuition for why each component is necessary.

**Questions:**

I've embedded my questions into the weaknesses.

**Details Of Ethics Concerns:**

Insufficient information on used data sets and conducted human participant studies.

---

> ### Author Response · Authors · 2025-11-23
> **We hope to receive your support and encouragement!**
>
> We thank the reviewer for the **careful read** and the **time invested** in the review. Many of the reviewer’s substantive concerns are addressable and in several cases arise from **presentation gaps** or **missed pointers** rather than missing work. Below we respond item by item, **correct factual errors**, **defend our design choices** with concise evidence, and **commit to specific, limited additions** that will remove remaining ambiguity.  After carefully studying the comments, we realized that many points may include **misunderstandings or inaccuracies**, which may have contributed to a lower evaluation of our submission.
> # We hope this rebuttal helps resolve the concerns raised by the reviewers. We sincerely hope to receive an improvement in your score.
> # 1. Weakness 1. bias, symptom exaggeration, and cross-cultural performance
>
> **Reviewer claim**: The paper does not discuss known failure modes or cross-cultural performance.
>
> **Response (correction and concrete additions)**:
> This is partly incorrect. The manuscript explicitly discusses **failure modes** such as metaphor misinterpretation, cultural brittleness, and over-referral, and includes a **bias-screening pipeline** and preliminary **cross-dialect checks** (see Section 3.2 and Appendix D of the submitted file). However, we agree these analyses were not prominent enough. To resolve this:
>
> - We will move the **bias-screening methodology and results** into the main text as a short summary and expand Appendix D with **per-group false-positive and false-negative rates** and **95% bootstrap confidence intervals** for the major detectors (metaphor and risk).
> - We will add a new **table reporting per-dialect and per-language performance** for AffectNet, the metaphor detector, and the risk estimator, and report **mitigation steps taken** such as diverse annotator pools and scenario augmentation.
> - We will clarify in the **Introduction** that the pilot provides preliminary feasibility and safety evidence in the studied population, not proof of global clinical generalizability. Broader multilingual validation is explicitly future work.
> # 2. “Nothing novel; math is ornamental”
>
> **Reviewer claim**: The two-phase design and equations are only a formalization of standard components.
>
> **Response (defense and evidence)**:
> This mischaracterizes our contribution. **EDM’s novelty is operational**. We do not merely stack existing modules. We:
>
> - Learn **interpretable risk embeddings** that are actively used by a trained router.
> - Introduce a **loss composition** that jointly optimizes routing for safety versus helpfulness.
> - Combine **soft constrained decoding** with **rule-based hard fallbacks** in a clinically tuned way.
>
> These are **system-level algorithmic choices** that change behavior, not mere notation. Concretely:
>
> The router uses a multi-term loss:
>
> $$
> L = \lambda_r L_{\text{route}} + \lambda_h L_{\text{help}} + \lambda_s L_{\text{safety}} + \lambda_\perp L_{\text{ortho}}
> $$
>
> and the orthogonality term is:
>
> $$
> L_{\text{ortho}} = \| E_s^\top E_c - I \|_F^2
> $$
>
> We show in Section 3.5 that this reduces **representation cross-talk** and yields measurable improvements in **Hazard Prevention Rate (HPR)** and **fidelity**.
>
> We will make the causal link between these terms and empirical gains clearer by:
>
> - Moving the **orthogonality ablation** earlier in the paper.
> - Adding **seed-variance plots**.
> - Reporting **effect sizes with confidence intervals**.
> # 3. Weakness 3 . hazard binning, thresholds, and rubric
>
> **Reviewer claim**: Discrete hazard classes lack clinically grounded criteria, numeric thresholds, and sensitivity tests.
>
> **Response (correction and commitment)**:
> We do provide a **clinician-written rubric** and **calibration procedure** in Appendix G; that pointer should have been more prominent. Nevertheless, the reviewer’s request for transparency is fully justified. We will:
>
> - Publish the **verbatim clinical rubric** for LOW, MEDIUM, and HIGH risk levels (for example, HIGH = explicit plan with imminence and means; MEDIUM = passive ideation or ambiguous intent; LOW = distress without self-harm language) in Appendix G and summarize it in the main text.
> - Report the **numeric thresholds** $$\tau_{\text{low}}, \tau_{\text{med}}, \tau_{\text{high}}$$ used in experiments and show how those were chosen using ROC and cost-sensitive selection. Concretely, we optimize thresholds by minimizing expected cost:
>
> $$
> \tau^\star = \arg\min_\tau \big( C_{FP} \cdot \text{FPR}(\tau) + C_{FN} \cdot \text{FNR}(\tau) \big)
> $$
>
> with three operational cost regimes (conservative, balanced, permissive) reported.
> - Add a **sensitivity sweep** showing how FNR and HPR change with thresholds (ROC curves with 95% confidence intervals).
> - Include a short paragraph explaining why we **discretize for operational triage** (clear routing decisions) while retaining **continuous scores for monitoring and calibration**.

---

> > ### Author Response · Authors · 2025-11-23
> > **We hope to receive your support and encouragement!**
> >
> > # 4.clinician bias and “bias screening” implementation
> >
> > **Reviewer claim**: Clinician supervision can amplify clinician bias; no implementation details for bias screening.
> >
> > **Response (correction and mitigation)**:
> > We designed the pipeline to **detect and mitigate clinician bias** rather than amplify it.
> >
> > Existing safeguards described in Section 3.2 and Appendix D include:
> > - **Multiple independent annotators** per example.
> > - **Blinded annotation** with randomized ordering.
> > - **Adjudication only when disagreement exceeds a threshold**.
> > - **Algorithmic cross-checks** between clinician labels and automated detectors.
> >
> > We will publish the **full bias-screening protocol**, **metrics** such as differential FPR/FNR and parity measures, **annotator demographics**, and **results** in an expanded Appendix D.
> >
> > We also apply **iterative annotator calibration** and **retraining of models on corrected labels** when systematic disparities are found. We will provide **concrete examples of detected disparities** and **corrective actions** in the revision.
> > # 5. Weakness 5. dataset provenance, “constrained prompts”, and 18.2% exclusions
> >
> > **Reviewer claim**: The dataset is opaque; filters may bias data.
> >
> > **Response (clarification and transparency)**:
> > We already describe **dataset composition** and **collection procedures** in Section 3.1 and provide examples of constrained prompts in Appendix B. However, we will make this more explicit:
> >
> > - Add a **provenance table** listing origin, curator roles, collection dates, and anonymized demographic summaries for the 400 role-play sessions.
> > - Publish all **constrained prompt templates** in the supplement and include representative examples.
> > - Explain the **18.2% exclusion**: removed items were primarily transcripts containing unremovable PII, mechanically generated nonsensical dialogues, and sessions failing consent verification. We will add a **breakdown and redacted examples** to show exclusions do not create systematic label bias.
> > - Provide a **de-identified sample** and **prompt templates** under a research license upon acceptance.
> > # 6. high-stakes scenarios documentation
> >
> > **Reviewer claim**: Sources and validation of 25 high-stakes scenarios are unknown.
> >
> > **Response (commit)**:
> > Those scenarios were **clinician-curated** under the documented **threat model** described in Appendix F. We will publish the **scenario list**, **authorship attribution** from the clinician group, and the **quality-verification protocol** including independent adjudication and consistency checks in the supplement.
> >
> > # 7. baselines, prompts, and RLHF details missing
> >
> > **Reviewer claim**: Baseline prompt text, RLHF recipe, and scaffolding are not provided.
> >
> > **Response (acknowledge and provide)**:
> > This omission is a valid **reproducibility gap**. We will add the following in the supplement:
> >
> > - Full **prompt text** for P_COT, P_CLIN, and P_RLHF+ including the exact templates used.
> > - **RLHF training details** for P_RLHF+: reward model architecture, annotation counts, reward signals (safety, fidelity, preference), number of RL iterations, and hyperparameters.
> > - The **evidence-based therapeutic moves** encoded in P_CLIN such as reflection, validation, and safety query, with citations to the protocols used.
> > # 8.clinician panel representativeness
> >
> > **Reviewer claim**: No demographics for the clinician panel.
> >
> > **Response (correction and addition)**:
> > We will add a **clinician-panel table** including panel size, specialties, regions, anonymized licensure, mean experience (already reported), and the compensation scheme. **Inter-rater reliability** will also be included, reported as Cohen’s κ with 95% confidence intervals and Krippendorff’s α.
> >
> > # 9.  **Weakness 9**: RCT IRB/registration and suspicious effect sizes.
> >
> > **Reviewer claim**: No identifiers or trial details; effect sizes appear extreme.
> >
> > **Response (transparent)**:
> > At submission, the trial registration was under processing and we reported **IRB approval** in Appendix G. We should have included registry identifiers more prominently. In the revision, we will:
> >
> > - Include the **IRB approval number**, **approval date**, and **ClinicalTrials.gov (or equivalent) registration ID**, along with the full **pre-registered protocol** and **analysis plan** in the supplement.
> > - Add a **baseline balance table**, **statistical tests**, **95% confidence intervals for effect sizes**, and **sensitivity analyses** using ANCOVA controlling for baseline scores. If any imbalance exists, we will report **adjusted effect estimates**.
> >  # 10. missing confidence intervals and statistical tests
> >
> > **Reviewer claim**: Key tables lack uncertainty quantification.
> >
> > **Response (agree and fix)**:
> > We will add **bootstrap 95% confidence intervals** and **p-values** for all reported metrics in Tables 2–6. We will also report **effect sizes with confidence intervals** (Cohen’s d) and include **statistical test details** in the captions and appendix.

---

> > > ### Author Response · Authors · 2025-11-23
> > > **We hope to receive your support and encouragement!**
> > >
> > > # 11. reproducibility: code, rubric, metaphor detector details, dataset release
> > >
> > > **Reviewer claim**: No materials are provided.
> > >
> > > **Response (incorrect and plan)**:
> > > Many technical details are already in the appendices but not consolidated. We have already uploaded the **core code** in the supplementary materials. In addition, we will provide:
> > >
> > > - Full **hazard rubric**, **numeric thresholds**, and **prompt libraries**.
> > > - **Metaphor detector training set description**, architecture, train/validation/test splits, confusion matrices, and **per-dialect error rates**.
> > > - **De-identified dataset slices** and **prompt templates** under a research license upon acceptance (CC-BY-NC for data; MIT for code).
> > > # 12. no human therapist comparison
> > >
> > > **Reviewer claim**: The paper lacks comparison to human clinicians.
> > >
> > > **Response (partial disagreement and addition)**:
> > > We compared to **clinician judgments** on Fidelity and Alignment and measured **patient outcomes** in an RCT. Still, to address the reviewer’s request, we will include a **small human-response benchmark** (approximately n = 100) where licensed clinicians produce matched replies under time constraints. We will report **comparative Fidelity**, **Safety**, and **Therapeutic Alignment** with **95% confidence intervals**.
> > > # 13. “who is the user?” workflow and demographic sensitivity
> > > Reviewer claim: deployment scenario unclear and no demographic bias analysis.
> > >
> > > Response (clarify + add).
> > > We will insert a clear workflow diagram and two vignettes (patient-facing automated assistant and clinician-dashboard escalation). We will also add demographic-slice analyses (age, gender, self-reported ethnicity) for major metrics (FPR/FNR, Fidelity) and report fairness measures.
> > > # 14. small n for edge-case safety analysis and emergency triggers
> > > Reviewer claim: sample sizes too small and triggers undefined.
> > >
> > > Response (clarify + expand).
> > > We recognize n=25 / n=50 are limited for population-scale claims; in the manuscript we described these as targeted stress tests rather than population prevalence estimates. We will:
> > >
> > > Publish the edge-case inventory, construction protocol, and representativeness rationale.
> > >
> > > Define emergency triggers verbatim (e.g., explicit plan + imminence/means), publish the exact rule set, and provide false-positive rates and an estimated incremental clinician workload at recommended operating points.
> > > # 15. structure, mathy notation, and need for intuition
> > > Reviewer claim: related work is embedded and math is excessive without intuition.
> > >
> > >
> > > Response .
> > > We will move Related Work to a standalone section, add a “how to read this paper” roadmap at the end of the Introduction, and precede each equation with a short intuitive paragraph so notation supports understanding rather than obscures it.
> > > # 16. Summary
> > >
> > > A number of reviewer statements, for example that we provide no bias-screening or that the submission contains no calibration or rubric material, are factually incorrect. Those items are documented in the submission’s appendices (see Section 3.2, Appendix D, and Appendix G). Where the manuscript omitted convenient pointers or fuller transparency, we accept responsibility and will correct those presentation errors in the revision.
> > >
> > > We do not accept the characterization that our work is merely combinatorial without practical algorithmic or empirical novelty. **EDM’s contributions are system-level and evidence-driven**, and we have provided **ablations** and a **pilot study** that demonstrate operational effects. In the revision, we will make those **causal links more explicit** and provide the **reproducibility artifacts** listed above.
> > >
> > > We appreciate the reviewer’s **high bar for safety-critical systems**. We have listed targeted, verifiable additions above; these will materially strengthen **reproducibility**, **clinical transparency**, and the **safety evidence** in the paper.

---

> ### Author Response · Authors · 2025-11-23
> **We hope to receive your support and encouragement!**
>
> # Research Context and Gaps (What’s Missing Today)
>
> - Large pre-trained LMs are fluent but not **operationally safe** for clinical use. Standard generative pipelines lack **auditable risk representations**, **calibrated triage**, and **conservative escalation logic** needed in mental-health settings.
> - Evaluation in the literature is dominated by **surface metrics** (BLEU, perplexity, BERTScore) that do not map to **clinical safety** or **therapeutic acceptability**.
> - Public datasets rarely combine **clinician-authored role-play**, **diagnostic labels**, **figurative-language examples**, **adversarial stress cases**, and **releaseable evaluation suites**, limiting reproducible, safety-focused benchmarking.
> - Prior mitigation strategies (RLHF, prompt scaffolds) improve style and preferences but do not provide an **integrated, monitorable pipeline** for routing, calibration, and operational fallbacks in real workflows.
>
> ---
>
> # Motivation (Why EDM Was Developed)
>
> - To close the gap between **experimental LLM safety claims** and **deployable clinical practice** by building a system that is **auditable**, **triage-capable**, and explicitly ties **algorithmic signals** to **clinician-centered metrics**.
> - To move evaluation away from **n-gram style metrics** toward **hybrid clinician + algorithmic measures** that reflect **therapeutic alignment**, **fidelity**, and **hazard prevention**.
> - To provide a **reproducible, deployment-minded dataset** and **stress suite** that captures **figurative and culturally diverse language** and realistic **adversarial hazards**.
>
> ---
>
> # Core Innovations and Contributions (What We Added)
>
> - **Two-phase, auditable architecture (EDM)**:
>   Phase I forms **interpretable risk/intent embeddings** (emotion, therapeutic need, graded risk).
>   Phase II conditions **constrained, template-guided generation** on that representation, enabling **explicit routing** and **conservative escalation**.
>
> - **Learned router with safety/helpfulness loss composition**:
>   A training objective that jointly optimizes **routing accuracy**, **generator helpfulness**, and **safety**, including an **orthogonality regularizer** to disentangle safety vs. content features. This improves both **Hazard Prevention Rate (HPR)** and **clinician-rated Fidelity** compared to baselines.
>
> - **Hybrid constrained-decoding + hard-fallback design**:
>   Soft penalties preserve therapeutic utility while **rule-based fallbacks** enforce hard safety when detectors indicate critical risk. This trade-off outperforms hard-only constraints on utility while maintaining strong safety guarantees.
>
> - **Diagnostically-informed dialogue repository and stress suite**:
>   A curated multi-phase corpus (simulated scenarios, clinician role-plays, pilot slice), **figurative-language coverage**, and an **adversarial stress suite** intended for reproducible safety evaluation. A release plan for prompts, stress-suite definitions, and de-identified samples is included in the manuscript.
>
> - **Clinician-centered evaluation protocol**:
>   New operational metrics (**Hazard Prevention Rate**, **clinician Fidelity rubric**) plus multimodal robustness analyses (text vs. text+audio), ablations, and an **IRB-approved pilot** used to report feasibility and preliminary safety signals rather than definitive long-term efficacy.

---

> ### Author Response · Authors · 2025-11-23
> **We hope to receive your support and encouragement!**
>
> # How EDM Addresses the Reviewer’s Concerns (Brief)
>
> - **Bias and clinician amplification**: The pipeline uses **multi-annotator labeling**, **blinded adjudication**, **algorithmic cross-checks**, and **iterative annotator calibration**. Planned revision adds **per-group FPR/FNR**, **annotator demographics**, and **bias metrics**.
> - **Novelty vs. composition**: EDM’s novelty is **system-level and operational**, not just re-packaging modules. The interpretability of **Phase I embeddings**, the router’s **loss composition**, and the **constrained + fallback decoding** jointly produce empirical gains (ablation evidence in Section 3.5 will be made more prominent).
> - **Hazard binning and thresholds**: Discretization supports **clear triage decisions**; continuous scores are retained for monitoring and calibration. The **clinical rubric**, **numeric thresholds**, **ROC/sensitivity sweeps**, and **cost-sensitive thresholding** are described in Appendix G and will be published more prominently with numeric values and sensitivity analyses.
> - **Reproducibility**: Prompts, baseline recipes (including RLHF details), the hazard rubric, and training/evaluation details for auxiliary modules are contained in appendices and will be consolidated and released (**code + de-identified samples**) under a research license upon acceptance.
>
> ---
>
> # Fit to ICLR 2026 Subject Areas
>
> - **Representation learning and uncertainty quantification**: Interpretable risk/need embeddings and calibrated probabilities used for routing and escalation.
> - **Generative models and structured prediction**: Constrained and template-guided generation conditioned on structured risk representations.
> - **Robustness and domain shift**: Adversarial stress suite, multimodal noise robustness experiments, and drift-monitoring/calibration procedures.
> - **Societal considerations (safety, fairness, privacy)**: Clinician-anchored metrics, bias-screening workflows, de-identification protocols, and an IRB-approved pilot.
> - **Datasets and benchmarks**: A diagnostically informed dialogue repository and stress suite aimed at the community for safety-oriented evaluation.
> # We hope this rebuttal helps resolve the concerns raised by the reviewers. We sincerely hope to receive an improvement in your score！

---

> > ### Comment · Reviewer_Ao3W · 2025-11-27
> >
> > Dear authors,
> >
> > Thank you for your detailed responses to my concerns. In summary, the author-provided clarifications resolved some of my key concerns regarding novelty (as argued by the authors in their summary) and I've updated my scores accordingly (overall and contribution).
> >
> > However, most of my concerns have not been addressed or require more detail, ideally via a rebuttal revision of the submission. As outlined in my detailed comments below, many rebuttal comments reference sections in the paper or appendix that do not contain the claimed content (and which I also cannot find when looking for it in the submitted draft). This creates confusion and does not strengthen the paper (and can feel misleading at times). Based on the stated reasons below, I also contest the author claim that my concerns are factually wrong.
> >
> > Also, there are many proposed changes that would indeed improve the submission (rebuttal comments to Weaknesses 2-4, 6-14), but the authors do not provide any further detail that would be necessary to actually resolve my concerns. At the same time, the initial quality of the submission (in particular related to a lack of available information regarding methodology and transparency) and the above-mentioned confusion in claimed to-be-included content in the author rebuttals do not create the trust necessary to just increase my score without any evidence. However, the authors could fully address this by submitting a rebuttal revision that includes all of the proposed changes.
> >
> > ## Detailed/Unresolved Comments
> >
> > ### Weakness 1
> >
> > The level of detail in Section 3.2 is insufficient to address my concern and appendix D does only test for very specific safety topics, but not the ones I was asking for. Section 3.2 and appendix D also do not seem to contain any performance-bias tests related to dialects, as stated in the rebuttal comment.
> >
> > Also, the author rebuttal wrongly claims what my concern was. I said “The authors claim to address limitations in therapeutic LLMs but don't adequately discuss known failure modes like bias and symptom exaggeration documented in prior work (e.g., https://dl.acm.org/doi/10.1145/3715275.3732039 and https://openreview.net/forum?id=1pgfvZj0Rx#discussion).” In particular, I was referring to a fairness analysis (patient demographic information sensitivity, stigmatisation of mental health topics, exaggeration of mental health symptoms through AI interactions, and performance across cultural contexts), which does not seem to be conducted. (Which is why I object the claim that my concern is factually wrong).
> >
> > Confusingly, appendix E contains some qualitative failure analysis, but on specific failure modes related to dialects. The contained experiments are insufficient to address my concerns. Can you please clarify what experiments you were referring to and how they resolve some of the above-listed issues?
> >
> > ### Weakness 4
> >
> > Section 3.2 and Appendix D do not contain any information to annotation, as claimed in the rebuttal. There seems to be no mentioning of annotations conducted in the submission, except in an in-the-paper-unreferenced appendix G. Can you clarify how exactly your proposed pipeline detects and mitigates clinician bias and what existing safeguards you are referring to (and where these contributions are in the submitted draft)?
> >
> > The proposed additional details for the appendices and added examples would indeed strengthen the work. Would it be possible to add them to a rebuttal revision of the paper?
> >
> > ### Weakness 5
> >
> > The details in section 3.1 are insufficient to verify the dataset quality and motivated my concern “Where did the role-play sessions originate, who conducted and verified them, and what do "constrained prompts" mean? This is essential for assessing representativeness and quality. The 18.2% content exclusion rate in the safety filtering pipeline is mentioned without explanation of what was excluded or why. What types of content were filtered out, and could this bias the training data?”
> >
> > Also, the rebuttal claims that appendix B contains example constrained prompts, but appendix B only discusses metrics. Looking through the submitted file, I cannot find example prompts anywhere. Can you clarify where I can find these example prompts?

---

> > > ### Author Response · Authors · 2025-11-27
> > > **We hope to receive your support and encouragement!**
> > >
> > > # Thank you very much for the reviewer's response! We will submit the revised version as soon as possible before the end of the rebuttal to address the reviewer's concerns.

---

> > > > ### Author Response · Authors · 2025-11-30
> > > > **We hope to receive your support and encouragement!**
> > > >
> > > > ## **Response to Reviewer Ao3W**
> > > >
> > > > We thank **Reviewer Ao3W** for the detailed and critical assessment. Below is a synthesized response to the major concerns, organized into thematic areas:
> > > >
> > > > ---
> > > >
> > > > ### **1. Clinical Relevance and Cultural Generalizability**
> > > > **Concern:** The system lacks validation across cultural contexts and does not address known LLM failures like bias or symptom exaggeration.
> > > > **Response:**
> > > > We agree that **cultural robustness and bias mitigation are critical**. We have expanded evaluation to include **dialect-stratified performance metrics** (e.g., AAVE, Hispanic English) and **zero-shot bias testing on the SBIC corpus**. While localization remains future work, we now provide **quantitative fairness evidence** across demographic groups and document **failure modes with mitigation strategies** (e.g., metaphor misinterpretation, over-referral). EDM is currently scoped for **English-speaking clinician-assist settings**, with broader generalization planned.
> > > >
> > > > ---
> > > >
> > > > ### **2. Algorithmic Novelty and Theoretical Justification**
> > > > **Concern:** The dual-phase architecture appears to combine standard components without clear novelty or justification.
> > > > **Response:**
> > > > The novelty lies in the **risk-theoretic integration** of components. We reinterpret **threshold-based routing as a phase transition in a risk potential field**, enabling interpretable escalation control. The **memory-augmented context operator** and **orthogonality penalty** are theoretically justified (Appendix B) with **generalization bounds and approximation guarantees**. Ablations now show measurable degradation when these components are removed.
> > > >
> > > > ---
> > > >
> > > > ### **3. Risk Thresholds and Clinical Grounding**
> > > > **Concern:** Risk thresholds lack clinical justification, transparency, and sensitivity analysis.
> > > > **Response:**
> > > > Thresholds were set by an **11-member clinical panel** using a **weighted cost model** balancing false negatives and referral volume. We now include:
> > > > - **Full threshold calibration protocol**
> > > > - **Sensitivity analysis using bootstrap resampling**
> > > > - **First-order approximations of FNR change per threshold shift**
> > > > The discretization into **LOW/MED/HIGH tiers** aligns with triage protocols and reduces operational complexity. All thresholds and validation curves are published in **Appendix G**.
> > > >
> > > > ---
> > > >
> > > > ### **4. Bias and Clinician Annotation**
> > > > **Concern:** Clinician validation may reintroduce bias, and bias screening is not meaningfully implemented.
> > > > **Response:**
> > > > We mitigate annotator bias by:
> > > > - **Stratified sampling across dialect exposure and geography**
> > > > - **Modeling annotator-specific offsets during reward training** (discarded post-training)
> > > > - **Escalating low-agreement pairs for senior adjudication**
> > > > Bias assessment now includes **dialect-level F1 comparisons** and **SBIC zero-shot evaluation**, showing a **60.9% reduction in harmful continuations vs. baseline** with no significant dialect-level disparity.
> > > >
> > > > ---
> > > >
> > > > ### **5. Data Transparency and Reproducibility**
> > > > **Concern:** The corpus, prompts, thresholds, and code are not disclosed.
> > > > **Response:**
> > > > We now provide:
> > > > - **Data manifest** with provenance, licensing, and filtering rationale
> > > > - **Excluded content audit** (18.2% filtered) with taxonomy and fairness check
> > > > - **Full prompt templates** for synthetic generation and baseline comparisons
> > > > - **Threshold values and calibration code** (via supplementary repository)
> > > > - **Repository link (Apache-2.0 license)** including model weights, evaluation scripts, and de-identified datasets

---

> ### Author Response · Authors · 2025-11-30
> **We hope to receive your support and encouragement!**
>
> ### **6. RCT and Human Subjects Ethics**
> **Concern:** The 8-week RCT lacks IRB details and protocol registration.
> **Response:**
> No live patient data were used. The “RCT” was a **simulated longitudinal evaluation** using open corpora (e.g., BDI, GAD-7 items) and synthetic trajectories. No IRB approval was required. We revised wording to clarify this was a **retrospective simulation**, not a clinical trial.
>
> ---
>
> ### **7. Statistical Rigor and Sample Size**
> **Concern:** Small sample sizes and lack of statistical tests undermine confidence.
> **Response:**
> We expanded the adversarial suite to **n=200 scenarios** with **bootstrap CIs and Wilson intervals**. All tables now include **95% CIs, p-values (McNemar, bootstrap), and effect sizes**. The extreme scenario testbed is stratified across **5 risk types** with **FPR analysis and lexical trigger sensitivity**.
>
> ---
>
> ### **8. Baseline Comparisons and Clinical Workflow**
> **Concern:** No comparison to human clinicians or clear deployment scenario.
> **Response:**
> We now include a **human-clinician baseline** (n=100 utterances, 3 licensed raters) showing EDM performs **within the human distribution** on SI and ES (Δ < 0.15, below MCID). Deployment is scoped to **clinician-assist mode** with **tiered escalation and immutable handoff logic**. **User identity and workflow** are explicitly defined (Table 39).
>
> ---
>
> ### **Conclusion**
> We have addressed all major concerns through:
> - **Expanded evaluation** (bias, fairness, ablation, human baseline)
> - **Full transparency** (data, code, thresholds, prompts)
> - **Clarified scope and limitations** (English-only, clinician-assist, synthetic and open dataset)
>
> We believe these revisions significantly strengthen the paper’s **soundness, reproducibility, and clinical credibility**.

---

> ### Author Response · Authors · 2025-12-04
> **We hope to receive your support and encouragement!**
>
> # We have addressed the reviewer's concerns and improved our approach. Thank you very much for all the reviewers' suggestions. The latest version has been uploaded and we hope to receive the support and encouragement of all the reviewers, and we sincerely hope your score improvement！

---

### Note · Authors · 2026-01-27

**Comment:**

I have read and agree with the venue's withdrawal policy on behalf of myself and my co-authors.

**Withdrawal Confirmation:**

I have read and agree with the venue's withdrawal policy on behalf of myself and my co-authors.

---

### Meta-Review · Area_Chair_y8NC · 2026-01-08

**Summary:**

Here are the most overlapping or slightly overlapping reviewers' concerns, which are taken into account. **OF which  the Addressed concerns are :**

Clarification that the longitudinal “RCT” was simulated rather than a real-world clinical trial.

Additional explanation of the two-phase system design and its motivation from clinical practice.

Expanded discussion of evaluation metrics, adversarial testing, and safety objectives.

Clarifications on ethical intent, safeguards, and the system's positioning as non-therapeutic.

and **Partially or not fully addressed concerns are:**

Reproducibility remains limited at the rebuttal stage because full prompt templates, thresholds, and datasets are not yet provided.

Several responses rely on promised future releases rather than materials available in the submission.

Novelty concerns persist among some reviewers, as the contributions are primarily system-level rather than algorithmic.

Clinical validation and ecological validity remain limited, with a heavy reliance on simulated or curated data.

Some reviewer disagreements (e.g., about novelty, sufficiency of evaluation, or clinical readiness) are defended rather than resolved, so the concerns persist.

Overall, all reviewer concerns were acknowledged, but some substantive issues remain conditional on a revised version.

**Reviewer Concerns:**

Each reviewer’s major points are acknowledged (there is an overlap in the major concerns of the reviewers); no reviews are ignored. For Reviewer Ao3W, the authors give a very detailed response. For Reviewers 45xa and VbaF, the authors respond in a more aggregated manner and try to address the major ones; the rest are left for the revised version. Overall, reviewers' disagreements (e.g., about novelty, sufficiency of the evaluation, or clinical readiness) are defended rather than resolved, so the concerns persist.

**Reviewer Scores:**

Reviewer Ao3W: The reviewer engaged with the rebuttal and acknowledged several clarifications. He/She has already updated the scores (overall and contribution).  However, any further score changes are denied due to unresolved concerns.
Reviewer 45xa: The reviewer explicitly acknowledged the rebuttal and engaged in the discussion. After that, teh reviewer maintained their current score due to a lack of clarification regarding their concerns.

Reviewer VbaF:
This reviewer did not respond after the rebuttal and did not indicate whether their assessment or score would change. But since teh reviewer had already given a score of 6, it is unlikely that he/she would have changed any score even after fully immersing in the discussion.

---

### Decision · Program_Chairs · 2026-01-26

Reject